# SARS-CoV-2 Infection Dynamics in the Pittsburgh Zoo Wild Felids with Two Viral Variants (Delta and Alpha) during the 2021–2022 Pandemic in the United States

**DOI:** 10.3390/ani13193094

**Published:** 2023-10-04

**Authors:** Deepanker Tewari, Ryan Miller, Julia Livengood, Leyi Wang, Mary Lea Killian, Felipe Bustamante, Candy Kessler, Nagaraja Thirumalapura, Karen Terio, Mia Torchetti, Kristina Lantz, Justin Rosenberg

**Affiliations:** 1Pennsylvania Veterinary Laboratory, Pennsylvania Department of Agriculture, Harrisburg, PA 17110, USAjullivengo@pa.gov (J.L.); fbustamant@pa.gov (F.B.); nthirumala@pa.gov (N.T.); 2Illinois Veterinary Diagnostic Laboratory, College of Veterinary Medicine, University of Illinois, Urbana, IL 61802, USA; leyiwang@illinois.edu; 3National Veterinary Services Laboratory, United States Department of Agriculture, Ames, IA 50010, USA; mary.l.killian@usda.gov (M.L.K.); mia.kim.torchetti@usda.gov (M.T.); kristina.lantz@usda.gov (K.L.); 4Zoological Pathology Program, University of Illinois, Brookfield, IL 60513, USA; kterio@illinois.edu; 5Pittsburgh Zoo and Aquarium, Pittsburgh, PA 15206, USA; jrosenberg@pittsburghzoo.org

**Keywords:** SARS-CoV-2, coronavirus, RRT-PCR, Felidae, zoo, lions, tigers and lynx

## Abstract

**Simple Summary:**

Severe acute respiratory syndrome coronavirus 2 (SARS-CoV-2), the causative agent of COVID-19 disease, has been reported to infect multiple animal species besides humans. In this study, SARS-CoV-2 infection with two viral variants; alpha and delta in two separate infection waves among the large cats housed in the Pittsburgh Zoo and Aquarium during the 2021–2022 pandemic is described. In March 2021, all but one of the lions exhibited clinical signs consistent with SARS-CoV-2 infection coinciding with the infection of an animal keeper. Viral shedding was detected during the infection phase. In December 2022, during the second phase, SARS-CoV-2 infection occurred in tigers and lynx along with a likely reinfection of lions. In the infected animals exhibiting clinical signs, the signs lasted up to 6 weeks and viral shedding in feces was variable. Virus genome sequencing indicated that the alpha variant and the delta variant were responsible for the first and second waves of infection in the zoo animals, respectively, and the viruses were closely related to variants circulating in human populations at the time of infection.

**Abstract:**

Severe acute respiratory syndrome coronavirus 2 (SARS-CoV-2) has been reported in multiple animal species besides humans. The goal of this study was to report clinical signs, infection progression, virus detection and antibody response in a group of wild felids housed in adjacent but neighboring areas at the Pittsburgh Zoo. Initially, five African lions (*Panthera leo krugeri*) housed together exhibited respiratory clinical signs with viral shedding in their feces in March of 2021 coinciding with infection of an animal keeper. During the second infection wave in December 2021, four Amur tigers (*Panthera tigris altaica*) and a Canadian lynx (*Lynx canadensis*) showed clinical signs and tested positive for viral RNA in feces. In infected animals, viral shedding in feces was variable lasting up to 5 weeks and clinical signs were observed for up to 4 weeks. Despite mounting an antibody response to initial exposure, lions exhibited respiratory clinical signs during the second infection wave, but none shed the virus in their feces. The lions were positive for alpha variant (B.1.1.7 lineage) during the first wave and the tiger and lynx were positive for delta variant (AY.25.1. lineage) during the second wave. The viruses recovered from felids were closely related to variants circulating in human populations at the time of the infection. Cheetahs (*Acinonyx jubatus*) in the park did not show either the clinical signs or the antibody response.

## 1. Introduction

Severe acute respiratory syndrome coronavirus 2 (SARS-CoV-2), the causative virus of the coronavirus disease 2019 (COVID-19), infection was first identified in late 2019 and is largely a disease of concern affecting human populations worldwide [1,2]. SARS-CoV-2 is an enveloped positive sense RNA virus that enters host cells via interaction between the viral spike glycoprotein and the host cellular angiotensin-converting enzyme 2 (ACE2) [3]. SARS-CoV-2 genetic sequences mutate periodically and give rise to variations of the original virus resulting in frequent emergence of variant strains [4,5,6]. As the virus has changed, different variants have become dominant at different times. The emergence of variant strains in the human populations are consistent with discovery of similar strains in animals, highlighting such events being linked to reverse zoonosis [7].

While the predominance of infections are in humans, SARS-CoV-2 infections have been shown to affect several animal species [8,9]. Farmed and wild animal infections with SARS-CoV-2, such as mink and deer have also been studied to assess the potential for these animals to serve as reservoirs for the SARS-CoV-2 virus infection in humans [9,10,11]. Companion animals, including domestic cats, dogs, hamsters, and ferrets can also be infected [12,13,14,15] as can wild animals in zoological parks [7,16,17,18,19]. Animals in zoological parks have received particular attention as these animals live in confined spaces, often in small cohort settings, sharing food and other resources and frequently have encounters with animal keepers and visiting public. 

In zoological park settings, several studies have shown that members of the Felidae family, particularly lions, tigers and snow leopards, can become infected. In April 2020, a Malayan tiger (*Panthera tigris jacksoni*) at the Bronx Zoo, New York City, USA tested positive along with co-housed animals, but with low reported transmissibility [7]. Lions also tested positive from about the same time period. Later, zoos in Barcelona (Spain), Uttar Pradesh and Rajasthan (India), Johannesburg (South Africa) and Indiana (USA) also reported SARS-CoV-2 infections in lions (*Panthera leo krugeri*), leopards (*Panthera uncia*) and pumas (*Puma concolor*) [16,20,21,22]. In addition, one study highlighted SARS-CoV-2 infection in animals belonging to the families Felidae, Viverridae and Procyonidae in a zoological parks despite vaccination [17]. 

In this report we describe the infection of lions, tigers and lynx at the Pittsburgh Zoo and Aquarium, USA during two different waves of human infection with animals showing signs consistent with SARS-CoV-2 infection. Infected animals were monitored for clinical signs and viral shedding in feces over the course of infection and reinfection of lions with suspected transmission of virus from animal keepers. The goal of the study was to understand infection exposure, disease progression, and viral shedding in a group of exposed and re-exposed animals showing clinical signs consistent with SARS-CoV-2 infection at the zoo.

## 2. Materials and Methods

### 2.1. Zoo Animals (Wild Felids)

Six African lions (*Panthera leo krugeri*) at the zoo were housed in an exhibit less than ~100 feet away, across the public walkway. Four Amur tigers (*Panthera tigris altaica*) and 5 Canadian lynx (*Lynx canadensis*) are housed in immediate proximity to each other. The only access point for lynx is to walk through the tiger area. The same keepers care for the lions, lynx, and tigers. Cheetahs (*Acinonyx jubatus*) are also located in same vicinity (Figure 1) and are also cared for by the same keepers. Upon animals testing positive for SARS CoV-2, biosecurity protocols were instituted to prevent further infections within the cohort or neighboring animal cohorts including caregiver isolation, and personal protective equipment used by the keepers. Other large wild felids that were housed in close proximity included 5 cheetahs in a separate exhibit and 2 Amur leopards. (*Panthera pardus orientalis*). Two new cheetahs were added in June 2022. The leopards were housed indoors next to lions separated by a walkway.

### 2.2. Samples

Between March 2021 and January 2022, a total of 193 fecal samples were collected for SARS-CoV-2 testing from lions, tigers, and lynxes that exhibited signs of upper respiratory infections consistent with SARS-CoV-2 infection as well as from animals living in the same cohort. Animal keepers initially collected 23 fecal samples from the two lions (Lion 1 and 5) exhibiting clinical signs and illness from 31 March to 21 May 2021 on at least weekly basis. Four other lions were not sampled or tested except for a one-time collection on 4 April 2021. Fecal samples were collected from the ground after animals had defecated. Caregivers used personal protective equipment including glove changes while collecting feces. To ascribe feces to a particular animal, caregivers only collected samples when they observed animal defecating. During subsequent illness in the second infection phase, 53 fecal samples were collected from 6 lions in the cohort from 21 December 2021, until 25 January 2022, at least once weekly. Fecal samples were also collected from 4 tigers (58 specimens) and 5 lynxes (55 specimens) from 1 December 2021, until 23 January 2022, and from 12 December 2021, until 23 January 2022, respectively, at least once weekly (Appendix A). Sera collected during routine health checks (17 sera; from 5 lions, 3 tigers and 7 cheetahs) were stored frozen and obtained from the sera bank for SARS-Cov-2 antibody testing. Amur leopards were not sampled as this group did not show any clinical signs or had any sera available for testing. Fecal samples were also not collected from cheetahs, but stored sera were available from the serum bank.

### 2.3. Real-Time Reverse Transcriptase (RRT)-PCR

Initial testing was performed at the University of Illinois Veterinary Diagnostic Laboratory. After the initial positive result, testing was performed at the Pennsylvania Veterinary Laboratory (PVL) from additional collection time points. The RRT-PCR test results for each animal initially found to be positive were confirmed by the National Veterinary Services Laboratory (NVSL).

RNA extraction and RRT-PCR were performed as previously described for animal samples [7,17]. Briefly, RNA samples were extracted on the KingFisher Flex using the MagMAX Pathogen RNA/DNA Kit. RRT-PCR was performed on Applied Biosystems™ 7500 Real-Time PCR Systems using AgPath-ID One-Step RT-PCR Kit (ThermoFisher, Waltham, MA, USA) and either CDC N1 or N2 primers and probes (both not typically applied for initial testing). The thermocycler conditions included one cycle of 48 °C for 10 min and 95 °C for 10 min and 40 cycles of 95 °C for 15 s and 60 °C for 45 s. NVSL confirmation was performed using N1 and N2 RRT-PCR. Testing at PVL was also performed using N1 RRT-PCR following CDC protocol as also previously described 9]. 

### 2.4. Sequencing and Bioinformatic Analysis

Whole-genome sequencing was performed at the NVSL as previously described [9,23]. Viral RNA was amplified by tiling PCR and libraries were prepared using the Nextera XT DNA Sample Preparation Kit according to manufacturer instructions (Illumina, San Diego, CA, USA). Sequencing was performed using the 500-cycle MiSeq Reagent Kit v2. Sequences were assembled using IRMA v.0.6.7 and DNAStar SeqMan NGen v.14.0.1. Additional sequencing was performed at the PVL using a targeted amplification method with four pairs of primers to amplify a portion of the SARS-CoV-2 spike protein gene with MiSeq using 6100 Artic kit (Eurofins, Louisville, KY, USA). Sequences were deposited into GISAID [24]. Reference SARS-CoV2 sequences were downloaded from GISAID. Sequence alignment and construction of phylogenetic tree with maximum likelihood were performed using MEGA 7.0.26 (megasoftware.net (accessed on 3 August 2022)).

For comparison, two human host COVID-19 viral samples from Pennsylvania were selected from GISAID (EPI_ISL IDs 1689985, 7834221). These human references were selected because collection dates (4 April and 6 December) were close to the dates of felid infections. Using the NextClade platform, phylogenetic trees of the human and big cat samples were generated to determine the relatedness of the viruses. Portions the SARS-CoV-2 spike protein gene from lynx samples were sequenced and Prokka was used to annotate partial sequences of the spike protein gene (https://gxy.io/GTN:T00168 (accessed on 3 August 2022)). Partial sequences of the spike protein genes from lynx samples were aligned to those from lion and tiger samples using the European Bioinformatics Institute (EBI) Clustal Omega platform (https://www.ebi.ac.uk/Tools/msa/clustalo/ (accessed on 3 August 2022)).

### 2.5. Serology

The cPass SARS-CoV-2 Neutralization Antibody Detection Kit (GenScript, Piscataway, NJ, USA) was used to demonstrate presence of neutralizing antibodies. The assay was performed according to the manufacturer’s protocol and as previously described [14]. Briefly, samples and manufacturer supplied controls were diluted 1:10 with dilution buffer and mixed with RBD-HRP. The diluted samples, and positive as well as negative controls were diluted with HRP-RBD at 1:1 and incubated at 37 °C for 30 min. After incubation, 100 μL of RDB-HRP neutralization reaction mixtures were added to a 96 well plate pre-coated with recombinant ACE2 protein. The plate was incubated for 15 min at 37 °C, the sample mixture removed, and wells were washed with provided wash buffer. After the addition of substrate, the reaction was stopped, and plates read at 450 nm immediately afterwards. Data were interpreted as a percentage reduction (%reduction) based on OD_450_ intensity. A manufacturer-recommended cut-off of ≥30% signal reduction was used to indicate the presence of anti-SARS-CoV-2 neutralizing antibodies.

## 3. Results

Lions were the first group of animals to exhibit clinical signs including coughing and sneezing after a zookeeper tested positive for SARS-CoV-2 in March 2021. Feces from Lion 1 were tested after coughing was noticed, and the animal was positive for SARS-CoV-2 by RRT-PCR. Following confirmation of SARS-CoV-2, all lions living in the same enclosure were monitored for clinical signs and fecal viral shedding by PCR testing (Figure 2a). Lion 1 exhibited prolonged coughing for approximately 6 weeks lasting until May 2021. Other Lions (Lion 2, 3, 4 and 5) housed in the same enclosure also showed respiratory signs including coughing and sneezing. The PCR threshold cycle (Ct) values for viral detection in the feces of lions 1 and 5 ranged between 23.8–34.2 over the infection period (Figure 2a). Lion 1 tested positive on 31 March and virus shedding was observed until 11 April. Lion 5 tested positive on 4 April and shedding continued until 10 May, more than a month after initial shedding was detected. Lions 2, 3, and 4 exhibited clinical signs but when tested on 4 April 2021 were negative for SARS-CoV-2 with fecal monitoring. Fecal samples were not collected for testing from Lions 2, 3 and 4, after SARS-CoV-2 was not detected by RRT-PCR on initial testing. None of the other animals including Lion 6, tigers, Amur leopards and lynx housed in the neighboring enclosures showed any clinical signs while lions 1–5 were sick during the first wave of infection from March to May 2021.

In December 2021, tigers were the first Felidae group that showed clinical signs and were also SARS-CoV-2 positive upon testing of the feces with the RRT-PCR. Tigers showed clinical signs starting in December 2021 through January 2022 (Figure 2b). The first SARS-CoV-2 positive result in the tigers was observed on 1 December, and the last positive result was the 30 December. The tigers exhibited a variety of clinical signs while sick, including coughing, sneezing, vomiting, wheezing, lethargy, ocular and nasal discharge, stertorous respirations, and decreased appetite. Coughing was not noticeable in tigers in January but shedding in one of the tigers (Tiger 4) was observed for another week (Appendix A).

In the neighboring enclosure, coinciding with the sickness in tigers, the lions again became sick between December 2021 to January 2022 (Figure 2c). The lions also showed respiratory signs. However, none of the lions had positive fecal RRT-PCR for SARS-COV-2. All but one of the 5 lynxes (Lynx 1–4) living in the neighboring enclosure at the zoo were also sick between December 2021 to January 2022 timeline with variable clinical signs (Figure 2d). One lynx (Lynx 5), that was housed separately did not exhibit any clinical signs. Only one lynx had positive SARS-CoV-2 PCR test (Lynx 1) during this time indicating infection. The other lynxes (Lynx 2, 3, and 4) in the same enclosure exhibited variable signs such as coughing, lethargy, decreased appetite and or diarrhea.

SARS-CoV-2 viral genomes recovered from Lion 5 (GISAID EPI_ISL_ 2928452) and Tiger 4 (GISAID EPI_ISL 8145733) were compared to SARS-CoV-2 variants circulating in human population in the region at the time of infection in the zoo animals using the NextClade resource. The virus recovered from lions in April 2021 belonged to PANGO lineage B.1.1.7, while the virus detected from human population at that time in the region was from a neighboring clade-B.1.429. Similarly, the virus detected from tiger belonged to the PANGO lineage AY.25.1, while the virus detected from the human population at that time in the region was identified as lineage AY.3.1 (Figure 3). The virus recovered from lion sample during the first infection wave (April–May 2021) was an Alpha SARS-CoV-2 variant, while the virus recovered from tigers in the second wave (December 2021) was a Delta variant. There were 23 amino acid (AA) substitutions in the virus recovered from lion samples, with 7 substitutions in the spike protein compared to virus variants recovered from humans during that time. Similarly, there were 38 AA substitutions in the virus recovered from tiger samples, with 7 substitutions in that spike protein compared to human variants recovered at that time (Appendix A).

We attempted to sequence the SARS-CoV-2 genome from the fecal sample collected from a lynx sample (Lynx 1) that tested positive for SARS-CoV-2 virus by PCR. While we were unable to construct a consensus full viral genome sequence, we were able to sequence portions of the viral genome from the sample. Sequence alignment indicated a high similarity of the SARS-CoV-2 spike protein gene from the lynx sample with the spike protein gene of SARS-CoV-2 recovered from the lion and tiger in the zoological park (Figure 4).

Serology to detect exposure to SARS-CoV-2 was conducted using C-pass neutralization assay. Sera available from two of the lions (Lions 1 and 5) collected in June 2019, prior to the COVID-19 pandemic, showed no reactivity to SARS-CoV-2. After the infection, all lions (Lions 1 and 5 in September 2021 and Lions 2, 3 and 6 in June 2022) showed strong antibody response, as measured by interpolated serum titers and with competitive inhibition titers using the surrogate virus neutralization assay (Figure 5). Lion 6 did not show any clinical signs among the group, also seroconverted showing presence of neutralizing antibody and exposure to the virus. Similarly, sera from three tigers collected post SARS-CoV-2 infection (Tigers, 1, 2 and 3 in June 2022), also showed strong seroreactivity. In contrast, none of the seven cheetahs housed in the zoological park during the two separate infection waves had any clinical signs and serum samples collected from August 2021 through June 2022 showed no sero-reactivity with surrogate virus neutralization assay (Figure 5).

## 4. Discussion

The SARS-CoV-2 pandemic has had an enormous impact on the human population, but infections in other mammals have also been reported including companion, farm, zoo, and wild animals. As of 7 July 2023, at least 32 animal species from 17 animal families belonging to 6 animal orders were reported positive for SARS-CoV-2 (Figure 6). While most animal species that contract infection develop clinical signs, many animal infections do not result in mortality [25]. The SARS-CoV-2 virus continues to evolve with many variants emerging across the world. The variants have been categorized as the variant of interest (VOI), variant of concern (VOC), and variant under monitoring (VUM). Variants are detected after careful analysis through epidemic intelligence, and rules-based genomic variant screening and other scientific analysis. Five main SARS-CoV-2 lineages have been designated as the VOC (alpha, beta, gamma, delta and omicron variants). VOCs have increased transmissibility compared to the original virus and have the potential for increasing disease severity. In addition, VOCs have been shown to exhibit resistance to vaccine-induced and infection-induced immune responses, and thus possess the ability to re-infect previously infected and recovered individuals [26]. During COVID-19 pandemic, infections with the alpha variant in humans resulted in more severe disease, increased risk of intensive care unit admission, and increased mortality compared to infections with delta variant [27]. Such information on the impact of virus variants on the severity of the disease in animals is lacking. The present study is unique as it traces the infection dynamics and neutralizing antibody response in a group of felids housed in adjacent but neighboring areas in the zoological park. The chronology of events in the study also indicated a likely reinfection of lions with a different variant of the virus resulting in clinical signs consistent with the SARS-CoV-2 infection. These lions were housed in a close-by setting to other felids that were also concurrently infected with the SARS-CoV-2.

Lions at the Pittsburgh Zoo were the first species that showed clinical signs after an animal keeper was found infected with SARS-CoV-2. Strict biosecurity protocols including but not limited to caregiver isolation, and personal protective equipment use by the keepers were soon implemented to avoid infection of other susceptible animal species in nearby enclosures or living areas. Once the lions exhibited signs of a respiratory infection, and tested positive for SARS-CoV-2, other animals living in the same enclosure and neighboring enclosures were subsequently monitored for SARS-CoV-2 infections for several months. Viral RNA shedding was detected for a considerable period and clinical signs in some animals were apparent for at least a month after infection was initially diagnosed. The PCR detection and subsequent sequencing of viral RNA from the feces of infected animals also confirmed that the animals had contracted the virus. Viral sequences suggest that the viral variants detected from zoo animals were contemporaneous to variants circulating in humans during the two infection waves and likely indicates humans as the source of infection despite the clade differences. Interestingly, the zoological park was closed to the public during the first wave infection wave (March-May) but was open to the public during the second wave in December 2021. An Alpha variant of SARS-CoV-2 was recovered from the lions during the first wave infection (March to May 2021) in the park. During the second wave of infection starting in December 2021, the tigers, lynxes, and lions all showed signs of infection, and fecal samples from these animals tested by fecal PCR assay over the observed time. The viral shedding was detected in several tigers and one lynx but none of the lions were positive for fecal viral shedding during this second round of infection at the zoo. Also, during the second phase of infection, as animals started getting sick, none of the keepers showed signs of illness with SARS-CoV-2 infection. It is highly likely the source of infection during the second phase was from asymptomatic shedding from humans. During the second infection wave, the tigers were found to be infected first followed with lynx. The lions appeared to have been concurrently infected as they started exhibiting similar signs as during first wave but did not show any detectable viral shedding [28]. However, whether lions were truly infected with SARS-CoV-2 but due to pre-existing natural immunity cleared the virus quickly or it was present at levels below the limit of detection of the assay is uncertain. It is also possible, though less likely, that the lions were suffering from another respiratory illness that was not detected or the infecting strain had poor predilection for gastro-intestinal tract in lions. The virus during the second wave of infection was determined to be a Delta variant upon sequencing. The infection of animals was also corroborated seroconversions in both lions and tigers where such specimens were available for analysis.

In the present study, it appeared that the cheetahs either did not get exposed to SARS CoV-2 or resisted infection. Similarly, although Amur leopards were closer to the lion enclosure, they also did not show any clinical signs. Lack of detectable neutralizing antibodies to SARS-CoV-2 in serum samples from cheetahs also corroborated absence of infection. Cheetahs are interesting animal group showing genetic monomorphism [29]. Cheetah have previously showed extreme susceptibility to other coronavirus infections that can result in high mortality, while other cats typically develop milder infections [30]. Thus far, reports from zoo animal infections from South Africa, Czech Republic, Sweden, Sri Lanka, India, Spain, and the United States have not reported SARS-CoV-2 infections in cheetahs but infections of other big cats including lions, leopards, tigers, and lynxes are well documented.

In most studies of SARS-CoV-2 infection in captive animals, evidence has pointed toward the caretakers as their primary source of transmission [7,16,31] and in some cases a definitive source was not identified. A more recent study has suggested transmission of SARS-CoV-2 infection from African lion to zoo keepers [22]. This is an area of concern as outlined in our study as mutations can occur when a virus infects a new species that do not serve as dead-end host. Other studies have also noted infection of large felids with distinct viral variants in different zoos worldwide [17,25,32]. The infection of new species can potentially create new niches as natural reservoirs for SARS-CoV-2. Further studies are needed to assess the impact of such mutations among variants towards a potential broadening of the susceptible host range.

In the present study, infections were cleared in all felids without any observed mortality, and the two variants resulted in similar clinical signs. The lions, tigers, and lynx cleared the infection and recovered within 4 weeks of illness. The infected animals showed strong antibody response as determined by a surrogate virus neutralization assay. These findings were similar to other studies in tiger and lions [7,16], where high titers of neutralizing antibodies against the viral spike protein, specifically the RBD domain of SARS-CoV-2, have been demonstrated. Interestingly, even though cheetahs and amur leopards in the zoo were being cared for initially by the same keepers neither showed any clinical signs nor seroconverted. During the second wave of infection in the zoo, lions with detectable antibodies did not show any viral shedding in feces but did show clinical signs consistent with SARS-CoV-2 infection. Considering humane, biosafety, and biosecurity reasons, sampling of upper respiratory tract was not attempted. However, collectively the findings of the study indicated that prior exposure likely will not prevent reinfection in lions but reduce viral shedding. The data presented in the current study provide strong evidence for the role of neutralizing antibody in reducing the severity of disease and viral shedding in felids. Therefore, vaccine mediated protection is likely to be beneficial in felids. Additional studies to understand the nature and duration of immunity in felids is likely going to be helpful for preventing infection in both human and animal populations.

## 5. Conclusions

In summary, we report SARS-CoV-2 infection dynamics in the large felids from the Pittsburgh Zoo. The animals were monitored for clinical signs and fecal shedding of virus and viral variants and antibody response. The viral variants belonging to Alpha and Delta lineages were confirmed, contemporaneous to those variants circulating in the humans at the time of infection.

## Figures and Tables

**Figure 1 animals-13-03094-f001:**
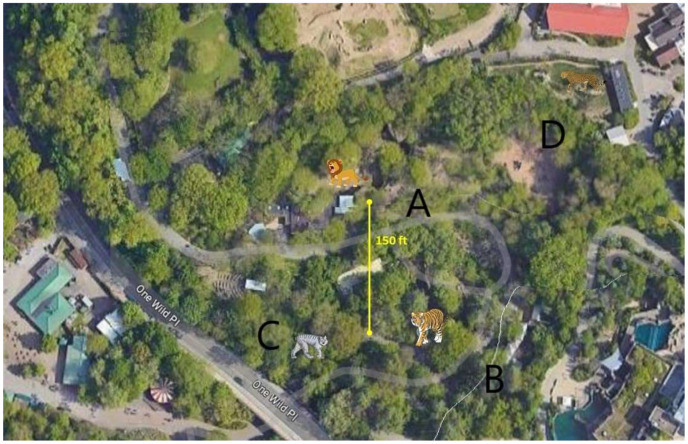
Aerial view of the zoological park showing the areas that house different felids including lions (A), tigers (B), lynxes (C) and cheetahs (D).

**Figure 2 animals-13-03094-f002:**
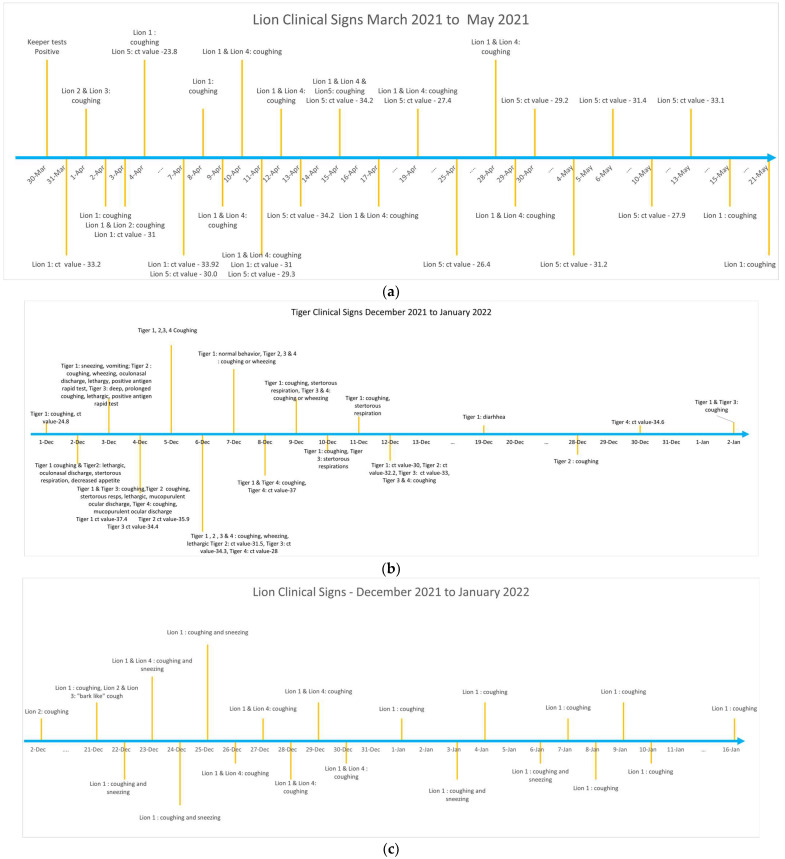
(**a**) Chronology of clinical signs and fecal shedding of SARS-CoV-2 in lions during the first wave of infection (March to May 2021) at the Pittsburgh Zoo and Aquarium. (**b**) Chronology of clinical signs and fecal shedding of SARS-CoV-2 in tigers during the second wave of infection (December 2021 through January 2022) at the Pittsburgh Zoo and Aquarium. (**c**) Chronology of clinical signs and fecal shedding of SARS-CoV-2 in lions during the second wave of infection (December 2021 through January 2022) at the Pittsburgh Zoo and Aquarium. (**d**) Chronology of clinical signs and fecal shedding of SARS-CoV-2 in lynx during the second wave of infection (December 2021 through January 2022) at the Pittsburgh Zoo and Aquarium.

**Figure 3 animals-13-03094-f003:**
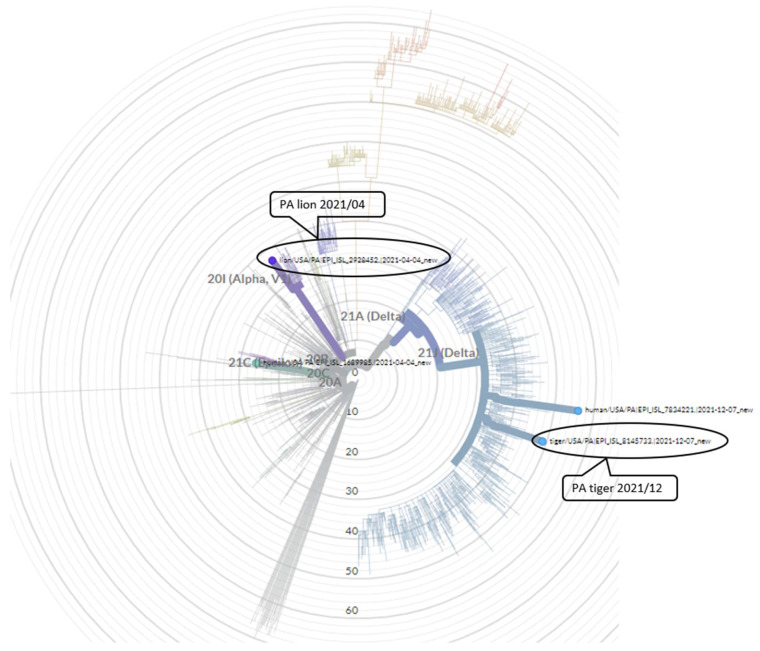
Circular phylogenetic tree generated by NextClade. The SARS-CoV-2 variant recovered from lion (depicted in purple) in April 2021 belongs to PANGO lineage B.1.1.7. The SARS-CoV-2 variant recovered from tiger (depicted in blue) in December 2021 belongs to PANGO lineage AY.25.1 showing similarity to other SARS-CoV-2 viruses.

**Figure 4 animals-13-03094-f004:**
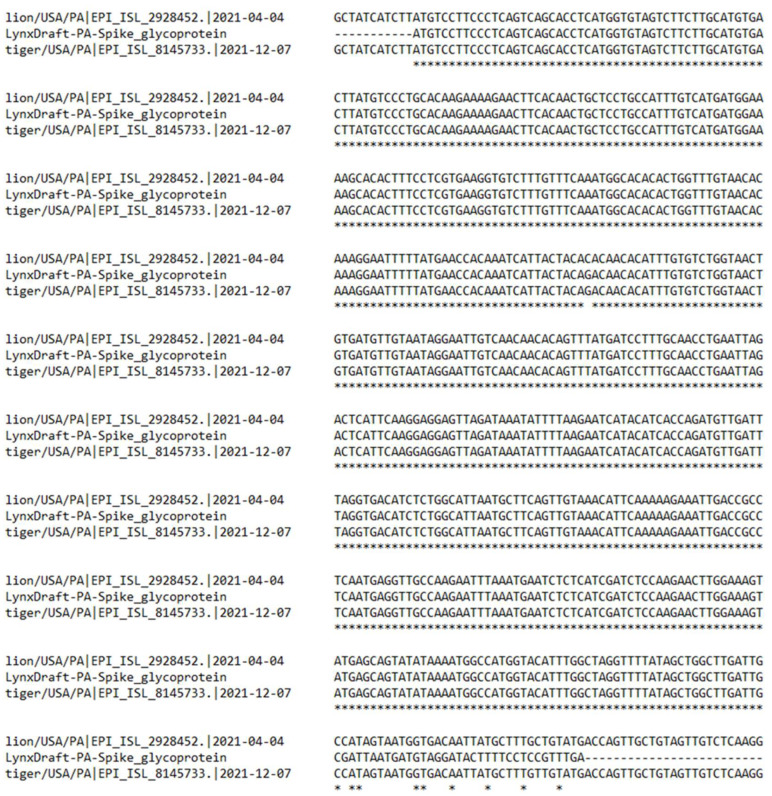
Partial sequence of the SARS-CoV-2 from lynx was aligned to a portion of the spike protein gene of SARS-CoV-2 recovered from the lions and tigers at the zoo. * depicts sequence homology.

**Figure 5 animals-13-03094-f005:**
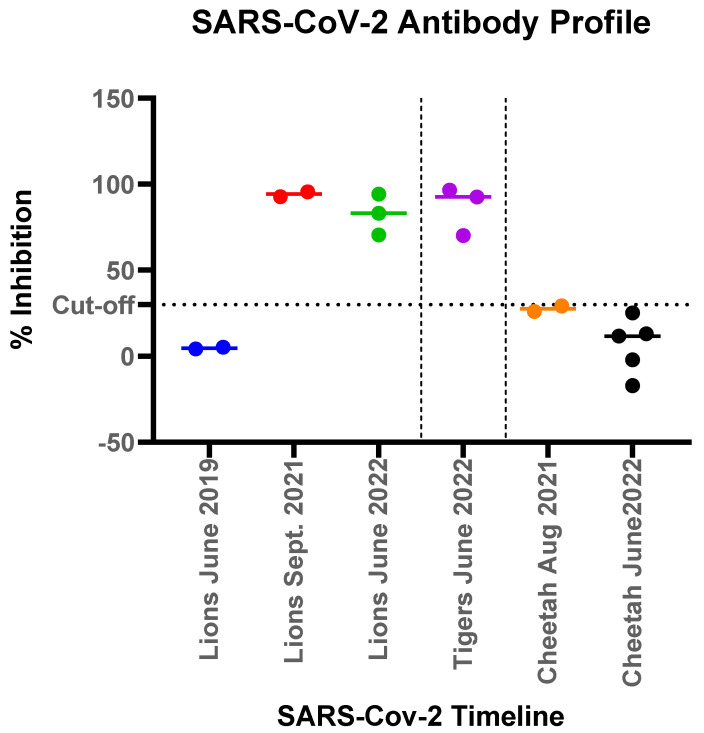
Antibody responses measured by a surrogate virus neutralization assay to SARS-CoV-2 in felids at the Pittsburgh Zoo. The % competitive inhibition for the C-pass assay observed for each of the animal groups (lions, tigers and cheetahs) was plotted to show pre-pandemic and post pandemic antibody response denoted by different colors for each time point. Lions tested in June 2019 and Sept 2019 were the same lions (lions 1 and 5).

**Figure 6 animals-13-03094-f006:**
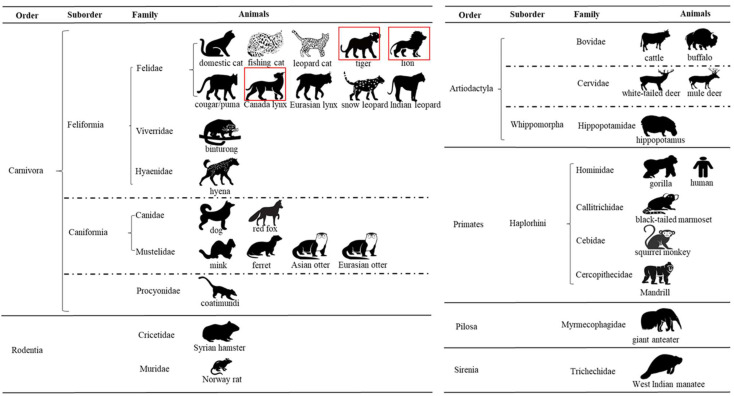
Current taxonomic diversity of animals detected with SARS-CoV-2. The diagram shows 32 animal species tested positive for SARS-CoV-2 within six animal orders: Carnivora, Rodentia, Artiodactyla, Primates, Pilosa, and Sirenia. Information on order, suborder, and family of these animal species is listed. In the present study, three animal species tested positive for SARS-CoV-2, and they are highlighted with a red color frame.

## Data Availability

Sequencing data of SARS-CoV-2 recovered from animals in the current study is deposited in the GISAID database.

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
