# Peer review of "SARS-CoV-2 Infection Dynamics in the Pittsburgh Zoo Wild Felids with Two Viral Variants (Delta and Alpha) during the 2021–2022 Pandemic in the United States"

_animals, 2023, doi:10.3390/ani13193094_

Round 1

Reviewer 1 Report

All my comments are in the enclosed file. 

Author Response

We thank reviewer 1 for very useful feedback to enhance the manuscript.

Here is point wise response to the concerns in the response

Main comment:
- Concerning paragraph 2.1, it would be useful to know how many lions, tigers and lynxes share the same areas. We only know that there is 2 Amour leopards (line 89). The lion n°6 is firstly mentioned in the results line 234. Concerning the lynxes, it is difficult to understand that there are 5 lynxes (as mentioned line 102) or two groups of lynxes, one with two lynxes (line191) and other one.

  • The Zoo animal section was updated and Fig 1 was also updated as suggested. Total 6 lions, 4 tigers and 5 lynx were part of the main study. Cheetahs (total 7 over the entire period) and Amur leopards (2) were also present but feces were not collected for cheetahs and leopards. Sera was available from serum bank for cheetahs.

  • - Concerning paragraph 2.2, may be a table with every sample (feces and blood) for each animal and date of sampling will be helpful. Actually, some information are present for species and not for the others.
    For example, it is written that feces lynxes are sampled once a week but we do not know for the other species. 
  • Sample section was rewritten and also Suppl. Table 3 was added for fecal collection dates and PCR results besides the positives being recorded in the figures.
    Why did you sample only two lions (lion 1 and lion 5) in March 2021 whereas lions 2,3 and 4 (at least) had clinical symptoms?
    Why did you not sample tiger 3 with clinical symptoms in December 2021?
    Why did you sample 5 lynxes whereas only four had clinical symptoms in December 2021?
  • Samples reported were the only samples available for analysis. The other samples were not available and or collected after initial analysis was negative for Lions 2, 3 and 4 but is stated under Samples section.
    Cheetah were mentioned in paragraph 2.1 but not in paragraph 2.2.
    - Concerning results, Fecal samples were not collected from cheetahs but sera were available from serum bank.
    L Lions 2, 3, 4 and 5? In this case, it is discordant from paragraph 2.2 (you sampled only lions 1 and 5)? - Lines 172- -CoVdiscordant from paragraph 2.2?
    What about Lion 6 in March 2021 and in December 2021?
    Lines 190-
  • This above issue is now clarified and matches Section 2.2
    in the same enclosure? Why there is no feces results about 5 lynxes (as described in paragraph 2.2?
  • Only virus detection results  are shown in the figure 2a, 2b 2c and 2d along with clinical signs to avoid confusion. As suggested we have also added other results in suppl. table.
    Lines 204 and 206, from which animal(s) and from which sample(s) did you recovered viral genomes? Also shown in Figure with Ct values but now stated  Line 231: from which lions did you collect sera in 2019? Now stated also in Suppl table. Thanks for the suggestion. Samples that are sequenced are from Lion 5 and Tiger 4. It is now stated. This is ascribed in GSAID database  GISAID EPI_ISL_ 2928452 and GISAID EPI_ISL 8145733.
    Line 232: It is not all lions but only two in September 2021 and three in June 2022. From which lions did you have seroconversion? Were they the same in September 2021 than in June 2022? When did you sample lion 6? Why the lion 6 is not on Figure 5? WE only had access to total 17 sera from serum bank. 7 sera samples were from 5 lions in the serum bank. Animals were not bled for this study and samples were stored in serum bank from routine health exams. Lions 1 and 5 were same in June and Sept.
    Line 236: from which tigers did you collect sera in August 2022? Now stated
    What about lynxes and Amur leopards serology? Not tested due to non availability and were not collected.
    Lines 284-285 about results of other animals living in the same enclosure and neighboring enclosures ... monitored for SARS-CoV-2infection for s
    leopards are described in results paragraph. We were limited by samples that were available for analysis.
    Minor comments:
    Line 65: Malayan tiger, please write the scientific name OK
    Line 68: lions, leopards and pumas, please write the scientific names OK
    Line306 to 309: May be there is a third hypothesis if we consider that the second variant delta have
    less or no digestive tropism in lions and only respiratory tropism. Liked the suggestion and now is included.

Reviewer 2 Report

These authors present what is for me one of the first SARS-CoV-2 original works in wildlife with a complete, well-designed and structured methodology, and not only an isolated case report where many hypotheses but few conclusions can be taken. Therefore, I believe authors should receive credit and recognition for their work.

I sincerely do not have many comments/corrections regarding this manuscript. I think it has interested to the readers; it is well-written and well-structured. However, I believe it would improve even more if the authors consider the following aspects:

- "many animal infections do not result in mortality as seen in humans during the early part of pandemic except in rare reports and specific species" (L252-253) - I do not agree at all with this sentence. The data available for animal species cannot be comparable to the data available for humans regarding COVID19, so the authors should definitely not compare mortality rates between animals and humans, because it lacks scientific data and evidence. Moreover, at a certain point, we could say that many human infections do not result in mortality in humans as well, of course depending on many factors (age, access to medical assistance...). I think the sentence is vague and suitable for ambiguous interpretations, so the authors should try to change it a little bit to avoid misinterpretations.

- I am not used to reading very informal expressions in scientific articles such as "We report here (...)". It is more likely to see "This study aims..." or even "We aim to" to express the goals of the study. I am not an English native speaker, as I believe the authors are. Therefore, I have to admit and respect if the authors do not want to change it. 

I wish the authors everything good.

Author Response

Thank you to the Reviewer 2 for a thoughtful review and your input to help improve the manuscript.

We have addressed the following issues

 "many animal infections do not result in mortality as seen in humans during the early part of pandemic except in rare reports and specific species" (L252-253) - I do not agree at all with this sentence. The data available for animal species cannot be comparable to the data available for humans regarding COVID19, so the authors should definitely not compare mortality rates between animals and humans, because it lacks scientific data and evidence. Moreover, at a certain point, we could say that many human infections do not result in mortality in humans as well, of course depending on many factors (age, access to medical assistance...). I think the sentence is vague and suitable for ambiguous interpretations, so the authors should try to change it a little bit to avoid misinterpretations. The sentence was removed to address ambiguity.

- I am not used to reading very informal expressions in scientific articles such as "We report here (...)". It is more likely to see "This study aims..." or even "We aim to" to express the goals of the study. I am not an English native speaker, as I believe the authors are. Therefore, I have to admit and respect if the authors do not want to change it.  Thanks and we have changed the informal expression based on this useful feedback through out the manuscript.

Reviewer 3 Report

Dear Editor,

many thanks for your invitation to review the paper entitled “SARS-CoV-2 infection dynamics in the Pittsburgh Zoo animals with two viral variants (Delta and Alpha) during the 2021-2022 pandemic in the United States” for “Animals” journal.

The manuscript by Tewari D. and colleagues on SARS-CoV-2 infection dynamics in wild felids in Pittsburgh Zoo (USA) presents an interesting piece of work in pointing out two viral variants (Delta and Alpha) during the 2021-2022 pandemic in the United States. The lions tested positive for alpha variant (B.1.1.7 lineage) and the tiger and lynx tested positive for delta variant (AY.25.1. lineage). The viruses recovered from large cat were closely related to variants circulating in human populations at the time of the infection. Very interesting that the cheetahs living in the Zoo did not show positivity for SARS-CoV-2. The manuscript contributes molecular and serological evidence of SARS-CoV-2 infections in wild felids, providing additional data to the expanding host range of these viruses and deepening the dynamics of infection in a zoological institution.

The manuscript is well-written, with appropriate methods, interpretations, conclusions and requires some smaller adaptations before it is suitable for publishing. Minor revision is therefore recommended (listed below).

Article Title: I would suggest replacing “Pittsburgh Zoo animals” with “Pittsburgh Zoo wild felids (or large cats)”

1. Introduction:

-Page 2 of 13, Line 65: delete “202,0,” and insert “2020,”.

-Page 2 of 13, Line 67: I suggest to arrange the sentence as follows: “Later, zoos in Barcelona (Spain), Uttar Pradesh and Rajasthan (India), Johannesburg (South Africa) and Indiana (USA) also reported SARS-CoV-2 infections in lions, leopards and pumas.”

2.Materials and Methods:

2.1 Zoo animals – please insert (wild felids)

-Page 2 of 13:

Figure 1: the definition of the image needs to be improved in order to be captured somewhere especially the little “icons” of wild felids. It may also be useful to associate a number or an alphabet letter to each icon bringing it in the figure caption (e.g. lions 1).

2.2 Samples

-Page 3 of 13:

-Please, describe briefly how the samples were collected and the biosecurity measures taken.

-I suggest inserting a summary Table of  total animals sampled with the following informations: felid specie, sample type and sample size.

-Page 3 of 13, Line 106: add “Amur” before “leopards”.

2.3 Real-time RT (RRT)-PCR: please correct with (rRT)

-Page 3 of 13, Line 108: please include in the text the name of laboratory that carried out the analysis at University of Illinois.

-Page 3 of 13, Line 114: please delete “R  RT-PCR” and insert “Real-Time RT-PCR (rRT-PCR)”

2.4 Sequencing and bioinformatic analysis

-Page 4 of 13, Lines 126-128: check the following sentence, it is not clear “Additional sequencing was performed at the PVL MiSeq using a targeted amplification method with four pairs of primers to amplify a portion of the SARS-CoV-2 spike protein gene”.

-Page 4 of 13, Lines 136-138: this sentence has been repeated, delete “Using the NextClade platform, phylogenetic trees of the human and big cat samples were generated to determine relatedness of viruses.”

2.5. Serology

-Page 4 of 13, Line 145: please rewrite as indicated below: “The cPass SARS-CoV-2 Neutralization Antibody Detection Kit (GenScript)…”

3. Results

-Page 5 of 13, line 176: Figure 2. a. caption: remove the double period stop at the end

-Page 5 of 13, Line 178: please delete “R  RT-PCR” and insert “rRT-PCR”

-Page 5 of 13, line 187: Figure 2. b. caption: remove the double period stop at the end

-Page 5 of 13, Line 190: please delete “RRT-PCR” and insert “rRT-PCR”

- Page 8 of 13, Line 234: check the reference to Figure 6, it is not correct

- Page 9 of 13, Line 246: Figure 5 caption: remove the double period stop at the end

References

-Page 13 of 13:

-line 454: Check the reference 32.: remove the capital letters from the title

-line 458: Check the reference 33. as indicated below: Mitchell P.K.; Martins M.; Reilly T.; Caserta L.C.; Anderson R.R.; Cronk B.D.; Murphy J.; Goodrich E.L.; Diel D.G. SARS-CoV-2 B.1.1.7 Variant Infection in Malayan Tigers, Virginia, USA. Emerg Infect Dis. 2021, 27(12), 3171-3173. doi: 10.3201/eid2712.211234.

English language and style are fine/minor spell check required.

Author Response

Thanks for this important feedback and your time in carefully reviewing our manuscript and offering helpful advice to improve it. We very much appreciate each of the issues you bring up. Our point-wise response is as below:

1. Introduction:

-Page 2 of 13, Line 65: delete “202,0,” and insert “2020,”. Addressed

-Page 2 of 13, Line 67: I suggest to arrange the sentence as follows: “Later, zoos in Barcelona (Spain), Uttar Pradesh and Rajasthan (India), Johannesburg (South Africa) and Indiana (USA) also reported SARS-CoV-2 infections in lions, leopards and pumas.” Addressed

2.Materials and Methods:

2.1 Zoo animals – please insert (wild felids) Addressed

-Page 2 of 13:

Figure 1: the definition of the image needs to be improved in order to be captured somewhere especially the little “icons” of wild felids. It may also be useful to associate a number or an alphabet letter to each icon bringing it in the figure caption (e.g. lions 1). Addressed

2.2 Samples

-Page 3 of 13:

-Please, describe briefly how the samples were collected and the biosecurity measures taken. Addressed by rewriting and adding a suppl table.

-I suggest inserting a summary Table of  total animals sampled with the following informations: felid specie, sample type and sample size. Addressed, only sample size was not available.

-Page 3 of 13, Line 106: add “Amur” before “leopards”. Addressed

2.3 Real-time RT (RRT)-PCR: please correct with (rRT)-WE made sure same terminology was used all across as RRT-PCR and put RRT-PCR under the keywords.

-Page 3 of 13, Line 108: please include in the text the name of laboratory that carried out the analysis at University of Illinois. Addressed

-Page 3 of 13, Line 114: please delete “R  RT-PCR” and insert “Real-Time RT-PCR (rRT-PCR)” Used RRT-PCR

2.4 Sequencing and bioinformatic analysis

-Page 4 of 13, Lines 126-128: check the following sentence, it is not clear “Additional sequencing was performed at the PVL MiSeq using a targeted amplification method with four pairs of primers to amplify a portion of the SARS-CoV-2 spike protein gene”. Addressed

-Page 4 of 13, Lines 136-138: this sentence has been repeated, delete “Using the NextClade platform, phylogenetic trees of the human and big cat samples were generated to determine relatedness of viruses.” Addressed

2.5. Serology

-Page 4 of 13, Line 145: please rewrite as indicated below: “The cPass SARS-CoV-2 Neutralization Antibody Detection Kit (GenScript)…”Addressed

3. Results

-Page 5 of 13, line 176: Figure 2. a. caption: remove the double period stop at the end Addressed

-Page 5 of 13, Line 178: please delete “R  RT-PCR” and insert “rRT-PCR” RRT-PCR is used

-Page 5 of 13, line 187: Figure 2. b. caption: remove the double period stop at the end Addressed

-Page 5 of 13, Line 190: please delete “RRT-PCR” and insert “rRT-PCR” RRT-PCR is used

- Page 8 of 13, Line 234: check the reference to Figure 6, it is not correct Yes, Addressed

- Page 9 of 13, Line 246: Figure 5 caption: remove the double period stop at the end Addressed

References

-Page 13 of 13:

-line 454: Check the reference 32.: remove the capital letters from the title Addressed

-line 458: Check the reference 33. as indicated below: Mitchell P.K.; Martins M.; Reilly T.; Caserta L.C.; Anderson R.R.; Cronk B.D.; Murphy J.; Goodrich E.L.; Diel D.G. SARS-CoV-2 B.1.1.7 Variant Infection in Malayan Tigers, Virginia, USA. Emerg Infect Dis2021, 27(12), 3171-3173. doi: 10.3201/eid2712.211234. Addressed

Round 2

Reviewer 1 Report

Thank you for taking into account my comments.